# ON THE EXISTENCE OF UNIVERSAL SIMULATORS OF ATTENTION

## ABSTRACT

Previous work on the learnability of transformers —focused on examining their ability to approximate specific algorithmic patterns through training —has largely been data-driven, offering only probabilistic rather than deterministic guarantees. Expressivity, on the contrary, has theoretically been explored to address the problems *computable* by such architecture. These results proved the Turing-completeness of transformers, investigated bounds focused on circuit complexity, and formal logic. Being at the crossroad between learnability and expressivity, the question remains: *can transformer architectures exactly simulate an arbitrary attention mechanism, or in particular, the underlying operations?* In this study, we investigate the transformer encoder's ability to simulate a vanilla attention mechanism. By constructing a universal simulator $\mathcal{U}$ composed of transformer encoders, we present algorithmic solutions to replicate attention outputs and the underlying elementary matrix and activation operations via RASP, a formal framework for transformer computation. We show the existence of an algorithmically achievable, data-agnostic solution, previously known to be approximated only by learning.

## 1 INTRODUCTION

The vast adoption of Language Models across diverse fields of study — whether in task-specific applications (Lin et al., 2022; Haruna et al., 2025; Consens et al., 2025) or theoretical verifications (Strobl et al., 2024b) — has underscored the remarkable success of attention-based transformers. These models have demonstrated the ability to *learn* from tasks and function as *simulators* of a broad range of computational architectures. While ongoing investigations seek to characterize the representational power of trained transformers from both statistical (in-context (Mroueh, 2023; Kim et al., 2024)) and computational perspective (Merrill et al., 2020; Liu et al., 2023; Merrill & Sabharwal, 2024), a fundamental question remains unanswered: *irrespective of the complexity class to which a transformer belongs, can a mechanism simulate attention itself using only interactions between vanilla transformers?* Specifically, we ask whether such a mechanism exists that emulates the functioning of a single-layer transformer encoder, given we have access to a system with transformers as the only computational model. As such, they can be solely characterized by their parameters. Throughout our discussion, we refer to the self-attention mechanism of the transformer 'encoder'.

To put the problem into perspective, we highlight that theoretical analyses of transformers often involve *hard* (*unique* or *average*) attentions. The language, PARITY=$\{w \in \{0,1\}^* \mid \#_1(w) = 0(\mathrm{mod}\,2)\}$ in particular, has been contextual in most of the investigations. Hahn (2020) pointed out the inability of transformers toward recognizing the language. Although the learnability of such transformers did not prove amenable (Bhattamishra et al., 2020a), Chiang & Cholak's way of overcoming the drawback marked the explicit construction of a multi-layer multi-head softmax attention transformer (SMAT). The language, $k$-PARITY $= \{w \in \{0,1\}^n \mid S \subset \{0,1,\ldots,n-1\}$ and $\sum_{i_j \in S} w_{i_j} = 0(\mathrm{mod}\,2)\}$ where $|S| = k \ll n$, has been shown single-layer multi-head SMAT-learnable by Han & Ghoshdastidar. To further detail the representational power, of such SMATs, we mention the task Match$_2 = \{(s_i, s_j) \mid (s_i + s_j) = 0(\mathrm{mod}\,p)\}$, proposed by Sanford et al. (2023), where $S = (s_1, s_2, \ldots, s_{|S|}) \in \{1, 2, \ldots, p\}^{|S|}\}$ and $p$ is very large. While the same can be solved using a single-layer single-head SMAT, Match$_3$, an extension with three variables does not follow suit, even with the multi-layer multi-head extension. Our construction consolidates both notions of attention (hard and soft) into a unified computational model (namely $\mathcal{U}$) capable

of performing any single-layer multi-head attention. As a stepping stone, our investigation simulates single-layer multi-head SMATs using (average) hard attention transformers, in line with Yang et al. (2024b). In particular, we use Restricted Access Sequence Processing (RASP) (Weiss et al., 2021), a formal, human-readable framework that models transformer computation with parallel, attention-driven processing while enforcing constraints such as fixed computation depth, element-wise operations, and pairwise token dependencies. Our construction of $\mathcal{U}$ utilizes RASP such that given a single-layer transformer-attention $T$ and an input $X$, the output (say, $T(X)$), becomes *exactly* equal to the output of $\mathcal{U}$ on receiving the pair $\langle T, X \rangle$ (see Figure 1). By explicitly formulating the transformations required to simulate matrix operations, including transposition, multiplication, and inversion within the constraints of a transformer, our work provides a novel foundation for understanding the representational capacity of self-attention. This marks an improvement over Giannou et al. (2023)'s construction, which relies on a computational framework that is not entirely transformer-based and amplifies input size e.g., transposing a $d \times d$ matrix requires an $d^2 \times d^2$ input.

Our construction analogizes the rationale à la Universal Turing Machine (UTM). Observe that a UTM $U$ accepts the encoded pair $\langle \hat{T}, w \rangle$ if and only if the Turing machine $\hat{T}$ accepts the word $w$. Inspired by the same, Kudlek (2012) explored the (non)-existence of such universal automata for some weaker classes of automata, such as finite and pushdown automata. This, in turn, motivates the inquiry into the simulation of other computational models. Analogously, our constructed transformer network, $\mathcal{U}$, when deemed a language recognizer, can either accept or reject depending upon whether the original transformer encoder $T$ accepts or rejects $X$. When viewed as a *transducer*, it can produce the same output as the original transformer encoder, $T$, on input $X$. It is rather natural to explore the idea of such self-simulation for architectures coupled with decoder attention, given the Turing-completeness (Pérez et al., 2021). Ours, in contrast, involves encoder-only architectures, which have further limited computational capabilities (Strobl et al., 2024b). As defined in Hao et al. (2022), the self-attention mechanism introduced by Vaswani et al. belongs to the category of *restricted* transformers. Our simulations will be confined to this class of transformers due to their ubiquitous influence.

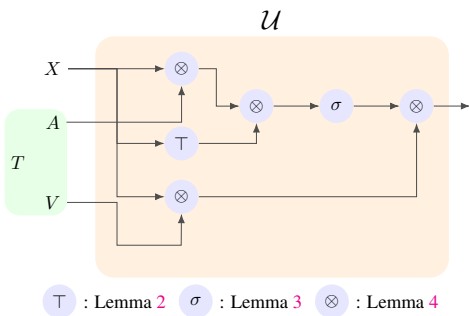

Figure 1: Simulation of attention $T$ characterized by matrices $A$ and $V$ on input $X$ using the proposed transformer network $\mathcal{U}$ such that $\mathcal{U}(\langle T, X \rangle) = T(X)$. Operations $\top, \otimes$ and $\sigma$ represent matrix *transposition*, *multiplication* and activation $\mathrm{softmax}$ implemented using transformer as presented in Lemma 2, Lemma 4 and Lemma 3 respectively.

Our approach also bridges a crucial gap between expressivity and learnability of transformer models. The problem $k$-PARITY, for example, achieves learnability through transformers (Han & Ghoshdastidar, 2025). On the other hand, our construction provides not only a definitive method to solve the same, but its applicability can also be generalized for related problems, e.g., Match$_2$. To contextualize, we point to the long line of works that explore the expressiveness of transformers in simulating important models of computation (Pérez et al., 2019; 2021; Hao et al., 2022; Barcelo et al., 2024), without determining the *exact* computational classes that include and are included by a transformer's recognition capacity. On the other hand, guarantees regarding the learning capacity of theoretically constructed transformers and their verification toward generalization onto the learned computation procedure (e.g., gradient descent in function space (Cheng et al., 2024); Newton's method updates in logistic regression (Giannou et al., 2025)) inherently become probabilistic and data-dependent. Such results lose justification in scenarios where approximation errors are unacceptable (e.g., formal verification). In contrast, our proofs provide a solution that *algorithmically enforces* correct attention behavior, ensuring reliability beyond data-driven approximations. From a probabilistic viewpoint, this can be regarded as an approximation guarantee with *certainty*, i.e., with $\mathbb{P}$-measure 1, given $X$ follows the law $\mathbb{P}$.

**Contributions.** The highlights of our study are as follows. **i)** We introduce a novel construction framework of amenable matrix operations underlying attention, such as transposition (Lemma 2), multiplication (Lemma 4), determinant calculation, and inversion (Lemma 9) using a transformer itself. We also show that algorithmic constructions exist that exactly represent activation outputs

(e.g., softmax (Lemma 3), MaxMin (Lemma 6)). The results combined present a new approach to proving a transformer encoder's expressivity towards a Lipschitz continuous function. Our constructions via RASP are available in the following repository `https://anonymous.4open.science/r/TMA`. **ii)** Our proposed simulator network $\mathcal{U}$ maintains parity with the architectures under simulation in terms of the following fundamental architectural resemblance. Due to its sole reliance on the number of input symbols, $\mathcal{U}$ possesses an inherent hierarchy while expressing attentions of increasing order, leading to a universal simulator (Theorem 5, 8, Corollary 5.2). **iii)** Given architectural specifications, our construction, for the first time, ensures the feasibility of simulating soft-attention (restricted transformer) using average-hard attention (RASP) (Remark 5.1, 8.1). The result extends to models involving multiple heads as long as their aggregation mechanism satisfies RASP-interpretability (Remark 5.2).

## 2 RELATED WORKS

**Simulation of computational models via transformers.** Existing work in this line lacks uniformity in transformer structures, leading to variations based on architectural distinctions (encoder-based or encoder-decoder) and the specific implementation of positional encoding. Introduced by Pérez et al. (2019) and Hahn (2020), a substantial body of research has investigated the theoretical capabilities of transformers, characterizing their expressivity in terms of diverse circuit families (Hao et al., 2022; Merrill et al., 2022; Chiang, 2025). Along this line of study, the development of domain-specific languages (DSLs) like RASP (Weiss et al., 2021), enabling the expression of self-attention and transformer operations in a human-interpretable manner, paved the way for further investigations (Zhou et al., 2024; Yang & Chiang, 2024; Yang et al., 2024a; Strobl et al., 2024a). Subsequently, RASP underwent refinements based on both augmentation and constraining of its features, leading to the creation of DSLs with enhanced expressiveness within their respective frameworks. Yang et al. (2024b) demonstrated the realization of *hard* attention through *soft* attention, involving the simulation of a logical language family that can be implemented by both mechanisms. Studies using a transformer as a language recognizer have also been pursued. Backed by the empirical studies Dehghani et al. (2019); Shi et al. (2022); Deletang et al. (2023), the expressivity of transformers has been investigated by measuring their equivalence with Turing machines (Bhattamishra et al., 2020b; Pérez et al., 2021). More recently, Merrill & Sabharwal (2023); Barcelo et al. (2024) have drawn equivalence with the logical expressions accepted by transformers. However, questions regarding the realization of the suggested solutions in a learning setup remain mostly open.

**Approximation and learnability.** While vanilla transformer encoders are, in general, universal approximators of continuous sequence-to-sequence (permutation equivariant) maps supported on a compact domain (Yun et al., 2020), they require careful construction to extend the property to models with nonlinear attention mechanisms (Alberti et al., 2023) and non-trivial positional encoding (Luo et al., 2022). However, it remains unclear whether the approximation capability holds while learning, given the unidentifiability of additional optimization errors due to data-driven training. In this context, we also mention that transformers are able to learn sparse Boolean functions of input samples having small bounded weight norms (Edelman et al., 2022). Along the line, Yau et al. (2024) ensures that multi-head linear encoders can be learned in polynomial time under $L^2$ loss. Corroborating Pérez et al. (2021)'s finding in a learning setup, Wei et al. (2022) also show that output classes of functions from TMs can be efficiently approximated using transformers (encoder-decoder). In contrast, the domain that has received the most attention is transformers' capacity to learn tasks in-context (IC) (Mroueh, 2023). Under varying assumptions on the architecture and data, transformers provably tend to emulate gradient updation (Ahn et al., 2023; Cheng et al., 2024), Newton's iterations (Giannou et al., 2025), and perform linear or functional regression (Fu et al., 2024; Pathak et al., 2024; Zhang et al., 2024). We reiterate that Giannou et al. (2023)'s OISC design augments inputs with scratchpad and memory, and outputs often include non-essential residuals (e.g., duplicate results of a matrix transposition, Lemma 19) unless post-processed. Even though their overall layer-count and head-count are constant, several hyperparameters lie intrinsically dependent on the number of tokens ($n$), e.g., the approximation bound is valid when temperature $\lambda \geq \log \frac{n^3}{\epsilon}$ (Lemma 2). Similarly, the assignment of the non-trivial parameters $V$ depends on $n$ (Lemma 20). Above all, its underlying computational framework fundamentally increases the depth (i.e., the layer-count) by allowing loops. Therefore, the ensuing computational power of the model becomes stronger compared to self-attention-based transformers (Hahn, 2020; Feng et al., 2023; Qiu et al., 2025). In

this work, we mitigate the limitations by answering whether transformer encoders can express with certainty the fundamental operations underlying the attention mechanism.

## 3 PRELIMINARIES

By *dimension* of a multi-dimensional array $M$, we signify the number of axes referred in $M$. Thus an element in $n$-dimensional array $M$ can be referred using the notation $M[i_0, i_1, \ldots, i_{n-1}]$. To reduce notational overhead, we denote by $M[i_0]$ the induced $(n-1)$-dimensional array hosted from the index $i_0$ of the introductory axis in $M$. Note that a matrix is a two-dimensional array. We also highlight the difference between the usage of $\odot_{i=1}^m (a_i)$ and $\circ_{i=1}^n (a_i)$. While the former, the concatenation operation, denotes the expression $a_1 \odot a_2 \odot \ldots \odot a_m$ as usual, the latter, an $n$-ary operation, is used to denote $\circ(a_1, a_2, \ldots, a_n)$. Given the sets of sequences $\mathcal{S}$ and $\mathcal{S}'$, we define a mapping $f : \mathcal{S} \to \mathcal{S}'$ as *length-preserving* if for any $S \in \mathcal{S}$, $|S| = |f(S)|$, where $|\cdot|$ implies the number of elements in the sequence.

### 3.1 TRANSFORMER ENCODER

A transformer encoder is a layered neural network that maps strings to strings. The input layer maps the string to a sequence of vectors. The subsequent layers apply the attention mechanism, which is composed of the sublayers' self-attention and feed-forward components. For ease of representation, we avoid the layer normalization mechanism. The final output layer maps the sequence of vectors back to a string. The following discussion formalizes the same.

**Input.** Let $\mathbf{w} = w_1 w_2 \ldots w_n$ be a string, where each character $w_i$ belongs to the input alphabet $\Sigma$. We assume the input layer of any transformer to be composed of the word embedding $\mathrm{WE} : \Sigma \to \mathbb{R}^d$ and positional embedding $\mathrm{PE} : (\mathbb{N} \times \mathbb{N}) \to \mathbb{R}^d$ in an additive form, so that the produced input vector becomes $X = (\mathbf{x}_1, \mathbf{x}_2, \ldots, \mathbf{x}_n) \in \mathbb{R}^{n \times d}$ such that $\mathbf{x}_i = \mathrm{WE}(w_i) + \mathrm{PE}(i, |w|)$.

**Encoder attention.** The first component of an encoder layer $\ell$ is self-attention. Assuming $X^{(0)} = X$, on an input $X^{(\ell-1)}, \ell \in \{1, 2, \ldots, L\}$ a self-attention mechanism produces

$$\sigma\left(X^{(\ell-1)} W_Q^{(\ell)} W_K^{(\ell)^\top} X^{(\ell-1)^\top}\right) X^{(\ell-1)} W_V^{(\ell)}, \tag{1}$$

where, $\sigma$ is a $\mathrm{softmax}$ activation, computing the attention scores from the query $X^{(\ell-1)} W_Q^{(\ell)}$ and key $X^{(\ell-1)} W_K^{(\ell)}$ to draw the influential value vectors $X^{(\ell-1)} W_V^{(\ell)}$ in a composite form, where the weight matrices $W_Q^{(\ell)}, W_K^{(\ell)} \in \mathbb{R}^{d \times d'}, W_V^{(\ell)} \in \mathbb{R}^{d \times d_v}$ and the input $X \in \mathbb{R}^{n \times d}$. A subsequent feed-forward layer, consisting of two linear transformations with a $\mathrm{ReLU}$ activation in between, is applied to this result. Note that, in the above expression the projections $W_Q^{(\ell)}$ and $W_K^{(\ell)}$ can be combined to result $A^{(\ell)} =: W_Q^{(\ell)} W_K^{(\ell)^\top} \in \mathbb{R}^{d \times d}$, and to simplify notations, we rename $W_V$ to $V$. A self-attention at a layer $\ell$ can thus be uniquely characterized by the three parameters $A^{(\ell)}, V^{(\ell)}$ and any normalizing activation function, here taken as $\mathrm{softmax}$. We will drop the notation $\ell$ wherever the context is self-explanatory. We call a transformer attention $T$ applied on input $X \in \mathbb{R}^{n \times d}$ of order $(n, d, d_v)$ if its characterizing matrices $A \in \mathbb{R}^{d \times d}$ and $V \in \mathbb{R}^{d \times d_v}$. Similarly, when a single-layer transformer $T$ with parameters $W_1 \in \mathbb{R}^{d_v \times d_1}$ and $W_2 \in \mathbb{R}^{d_1 \times d_2}$ in feed-forward sublayer is applied on input $X \in \mathbb{R}^{n \times d}$, we call it of order $(n, d, d_v, d_1, d_2)$.

### 3.2 GAHAT

A generalized attention, as proposed by Hao et al. (2022), takes the query and key as input and does not restrict them to be combined using the dot-product operation only. Instead, any computable association can be employed to calculate attention scores. Finding the dominant value vectors has also been kept flexible using a function $\mathrm{Pool}$ that takes the value vectors and the attention scores. When this function is particularly *unique* (or, *average*) hard, such transformers are regarded as generalized unique (or, average) hard attention transformers (GUHAT or GAHAT). As such, given value vectors $XV = (\mathbf{y}_0, \mathbf{y}_1, \ldots, \mathbf{y}_{n-1})$ and attention scores $(a_0, a_1, \ldots, a_{n-1})$, let $j_0, j_1, \ldots, j_{m-1} \in \{0, 1, \ldots, n-1\}$ are the indices in ascending order such that they maximize $a_j$s. Then, unique hard attention pools $\mathbf{y}_{j_0}$ while average hard attention pools $\frac{1}{m} \sum_{i=0}^{m-1} \mathbf{y}_{j_i}$.

The computational model underlying the Restricted Access Sequence Processing Language (RASP), introduced by Weiss et al. (2021), resembles that of GAHAT, based on overlapping sufficiency

characterizations (Section 3.1 in Weiss et al. (2021) and Section 4.3 in Hao et al. (2022) (Also, in the same light, the expression above (1) falls under the category of *restricted* transformers.)). RASP is a human-interpretable, sequence-processing DSL for designing transformer encoders. It operates on a sequence of tokens (e.g., characters, numbers, Booleans) to produce a length-preserved output sequence. Its core syntax includes elementwise operations and two non-elementwise operations: **select** and **aggregate**, which together correspond to a single self-attention layer. Token values and positions are accessed via **tokens** and **indices**. Lacking loops, RASP execution is inherently parallelizable operations, mirroring self-attention (via **select** and **aggregate** pair that resembles the $QKV$ operation), with elementwise operations reflecting terminal feed-forward layers. This absence of iterative constructs limits its applicability to inherently sequential computations, a direct consequence of the transformer's constant-depth nature that prevents arbitrary iteration simulation in one pass. Note that the **aggregate** operation is crucial for derived operations like **length**, which returns a sequence of the scalar repeated to maintain length.

Given the definition of Average Hard Attention (AHA) by Hao et al. (2022) (Def. 9), and the fact that **aggregate** performs an *average* over value vectors from the Boolean attention matrix generated by **select**, it is evident that the attention module in RASP is AHA. The **select** operation uses a Boolean predicate to associate keys and queries, placing it under the category of Generalized Average Hard Attention (GAHA). While GAHAT allows any terminating aggregator function[1] (which is a ReLU-activated FFN for restricted transformers), RASP permits any FFN for the same. The only sufficient condition for an activation to be compliant with RASP is universal (also uniform) approximation with arbitrary accuracy of regular maps, e.g., continuous Borel-measurable functions, Besov functions, etc.

In the scope of restricted transformers, various attention mechanisms have been employed to achieve faster computation, differing mainly in their choice of characterizing matrices and/or the Pool function. For instance, Linformer (Wang et al., 2020) is one that introduces new characterizing matrices $E, F \in \mathbb{R}^{k \times n}$ for some $k < n$ such that the attention becomes

$$\sigma\left((XW_Q)(EXW_K)^\top\right) FXV. \tag{2}$$

A linear attention (Katharopoulos et al., 2020), on the other hand, assumes no Pool, resulting in:

$$(XW_Q)(XW_K)^\top XV. \tag{3}$$

## 4 ON SIMULATING ATTENTION

In this section, we will provide all necessary lemmas and propositions required to construct the transformer network $\mathcal{U}$ simulating arbitrary transformer attention $T$ of order $(n, d, d_v)$ (Theorem 5). The first proposition presents a way to rearrange a multi-dimensional array to a single dimension so that all elements can be effectively accessed (proof in Appendix A.1). Subsequently, Lemma 2-4 use this representation to perform some basic operations such as matrix transposition, applying softmax activation and matrix multiplication[2]. Pseudocodes Algorithm 1-3 serve the constructive proofs of the respective lemmas via GAHAT. Listing 1-3 provide the corresponding RASP codes. Notice that, while these pseudocodes involve notation $r$ denoting the matrix order (i.e. the number of rows), we have considered $r = 3$ in the RASP codes as presented in Appendix A.1.

**Proposition 1.** *An $n$-dimensional array $A$ having size $m_l$ for each dimension $l \in \{0, 1, \ldots, n-1\}$ can be represented using a one-dimensional array $A'$.*

**Lemma 2.** *There exists a transformer transposing any matrix $A$ of order $r$.*

Note that for square matrices, the Algorithm 1 does not require the order $r$ explicitly. As RASP allows any arithmetic computation, $r$ can be determined from the expression $r^2 = $ **length**.

**Lemma 3.** *There exists a transformer implementing the operation softmax on matrix $A$ of order $r$.*

**Lemma 4.** *There exists a transformer multiplying matrices $A$ and $B$ of shape $r \times k$ and $k \times c$, for any $k \geq \frac{rc}{r+c}$.*

---

[1]The choice of the codomain of $g$ as $\{0, 1\}$ is purely based on the objective of language recognition.

[2]However, for notational convenience, the usual matrix indexing will be followed in these pseudocodes.

---

**Algorithm 1:** Transposing a matrix $A$ of order $r$.           [see Listing 1]

---

1   $r \hookleftarrow$ order of $A$.

2   Create a permutation $\rho$ of the indices of $A$, such that it maps an element $A[i, j]$ to position $(j, i)$ using the value $r$.

3   Create an attention that maps the indices of $A$ to the reflected indices obtained from the calculation of $\rho$.

4   Return $A^{\top}$ produced from passing $A$ to the attention matrix created above.

---

**Algorithm 2:** Applying $\mathrm{softmax}$ on a matrix $A$ of order $r$.         [see Listing 2]

---

1   $r \hookleftarrow$ order of $A$. Count the columns in $A$ as $c$.

2   Exponentiate all the terms in $A$, say to $A'$.

3   Create $r$ attention matrices each drawing the length-preserved sequence $A'[i]$ padded with 0, where $i \in \{0, 1, \ldots, r-1\}$.

4   Let $\mathrm{sum}$ denote the sequence such that $\mathrm{sum}[k] = \sum_j A'[i, j]$ for all $ci \leq k < c(i+1)$.

5   Return the resultant sequence $A' / \mathrm{sum}$.

---

**Algorithm 3:** Multiplication of matrices $A$ and $B$ of shape $r \times \cdot$ and $\cdot \times c$ respectively.   [see Listing 3]

---

1   Let $r$ (and $c$) $\hookleftarrow$ order of $A$ (and $B^{\top}$).

2   Create $r$ attention matrices each drawing the length-preserved sequence $A[i]$ padded with 0, where $i \in \{0, 1, \ldots, r-1\}$.

3   Similarly, create $c$ attention matrices each drawing the length-preserved sequence $B[:, j]$[3] padded with 0, where $j \in \{0, 1, \ldots, c-1\}$.

4   Multiply tokens from each row of $A$ with that of each column of $B$ and store the $rc$ sequences in $rc$ variables.

5   Create $rc$ attention matrices such that attention matrix $i$ focuses on first $\cdot$ positions of $i^{\text{th}}$ row.

6   Combine the sequences from line 4 with the attention matrices produced from line 5 to get $AB$, where the last $\cdot(r+c) - rc$ tokens are 0.

---

To address the issue of redundant tokens (from Algorithm 3) occurring consecutively at the sequence's end, we can incorporate a trivial attention mechanism in conjunction with a feed-forward network. This approach enables the contraction of a sequence with $m$ tokens into a shorter sequence of length $n$ (where $n < m$). To achieve this, a weight matrix $W^{m \times n}$ is employed within the final feed-forward sublayer such that $W[i][j] = 1$ when $i \leq n$, and $i = j$ and 0 otherwise. Having implemented the fundamental operations with transformers, we now present our main result.

**Theorem 5.** *There exists a transformer network $\mathcal{U}$ that, on any input $X$ of shape $(n \times d)$, can simulate any single-layer transformer attention $T$ of order $(n, d, d_v)$.*

*Proof.* Suppose the restricted transformer attention $T$ is characterized by $A$ and $V$ such that it can be expressed as $\sigma\left(X A X^{\top}\right) X V$. The network $\mathcal{U}$ simulating $T$ on input $X$ takes input $X$, $A$ and $V$; and it can be constructed through a series of fundamental operations, each of which has been realized by specific transformer architectures as mentioned in Algorithm 1-3. Figure 1 depicts the required network $\mathcal{U}$.        □

Note that the criteria in Lemma 4 can be satisfied by $k \geq \min(r, c)$. In the context of Theorem 5, this requires $n \leq \min(d, d_v)$. This not only aligns with the existing empirical scenarios where the sequence length $n$ is smaller than the representation dimensionality $d$ (Vaswani et al., 2017), but it also renders the relation between hidden dimensions immaterial. Additionally, the representational dimensions ($d$ and $d_v$) are often considered equal. In such scenarios, given that $n = 0 \pmod 4$, the construction of $\mathcal{U}$ becomes entirely dependent on the sequence length, i.e., the number of input symbols. The reason being, given the RASP primitive **length**, the provision to perform any arithmetic operation, and the value of $n$, we may deduce the value $d$. For example, to know the value of $d$ while multiplying $X$ and $A$, we may evaluate the expression $nd + d^2 =$ **length**.

**Corollary 5.1.** *There exists a transformer network $\mathcal{U}$ that can simulate any single-layer transformer encoder $T$ of order $(n, d, d_v, d_1, d_2)$.*

---

[3]The notation indicates all elements stored in column $j$ of matrix $B$.

*Proof.* The characterizing parameters of a transformer encoder from the class of restricted transformer contain two additional matrices. Let $W_1$ and $W_2$ specify the linear projections within the feed-forward sublayer, in addition to the attention and value matrices characterizing $T$'s self-attention sublayer. That is $T(X) = \mathrm{FFN}\left(\sigma\left(XAX^\top\right)XV\right)$ where, $\mathrm{FFN}(X) = \mathrm{ReLU}\left(XW_1\right)W_2$.

We can implement the operation $\mathrm{ReLU}$ (see Listing 4). Now, continuing from Theorem 5, the rest of the operations can be simulated using an application of Algorithm 3. □

**Remark 5.1** *(Simulating SMAT using AHAT).* The theorem ensures the existence of a unified network capable of simulating certain computational models while maintaining parity with the models under simulation. Precisely, we have employed average hard attention (see GAHAT in Section 3) to mimic $\mathrm{softmax}$-activated attentions. As a consequence, we can construct an AHAT for problems such as $\mathrm{Match}_2$, known until now to be only learnable using single-layer single-head SMATs.

**Remark 5.2** *(Simulating Linformer and Linear Attention).* As long as the characterizing matrices of the transformers are involved with matrix multiplication (e.g., Linformer (2)) and the function $\mathrm{Pool}$ is implementable using RASP (e.g., linear attention (3)), the Theorem 5 and Corollary 5.1 can be applied to achieve a transformer network $\mathcal{U}$ simulating them.

Remarkably, one may follow an alternative approach to proving the representational capacity of $\mathcal{U}$ by showing that it realizes operations such as (4) (see Appendix). The proof involves altering the construction of $\mathcal{U}$ by introducing final attention parameters that adapt to the input $\langle T, X \rangle$. It is crucial since, in the process, we show the existence of a transformer that inverts non-singular matrices of fixed orders. See Appendix A.3 for a contextual discussion.

## 5 DISCUSSION ON GENERALIZATION

Let us first analyze the complexity of the constructions given above. We define the *width* of a single encoder layer as the count of attention heads it contains. To extend this definition to multi-layer encoders, we define the width as the maximum width among all its constituent single-layer encoders. The shortcoming that makes the Algorithm 3 lengthy stems from explicitly mentioning the $r + c + rc$ variables. Even with classical implementation of matrix multiplication, where $C[i_0, i_1] = \sum_{i=1}^{k} A[i_0, i]B[i, i_1]$, taking $O(rkc)$ time, it does not resolve the issue, but

| Operation | Input Dependency | Cost |
|---|:---:|---|
| Transposition | ✓ | $O(1)$ |
| softmax | ✓ | $O(r)$ |
| Multiplication | ✓ | $O(rc)$ |
| MinMax | ✗ | $O(1)$ |

Table 1: The computational cost of construction associated with the operations and whether they are dependent on the order of the input matrices.

rather follows the same in the scope of variable renaming facility. In contrast, since the constructed system assumes attention being one of the basic operations and thus is an $O(1)$ operation, matrix multiplication costs $O(2rc)$ number of operations. Similarly, the computation cost for Algorithm 2 and Algorithm 1 for an order-$r$ matrix is $O(r)$ and $O(1)$, respectively. For each algorithm, the construction of the transformers ensures that their depth is not a function of the input; however, for most cases, the width *is* — a comprehensive view has been presented in Table 1.

**Corollary 5.2.** *Let $\mathcal{U}_{(n,d,d_v)}$ and $\mathcal{U}_{(m,e,e_v)}$ be transformer networks as defined in Theorem 5, where i) $\mathcal{U}_{(n,d,d_v)}$ simulates single-layer transformers with attention matrices $A \in \mathbb{R}^{d \times d}$, value matrices $V \in \mathbb{R}^{d \times d_v}$, and inputs $X \in \mathbb{R}^{n \times d}$, ii) $\mathcal{U}_{(m,e,e_v)}$ is defined analogously for dimensions $m$, $e$, and $e_v$. If $n \geq m$, $d \geq e$, and $d_v \geq e_v$, then $\mathcal{U}_{(n,d,d_v)}$ is at least as expressive as $\mathcal{U}_{(m,e,e_v)}$. Specifically, any computation performed by $\mathcal{U}_{(m,e,e_v)}$ can be exactly simulated by $\mathcal{U}_{(n,d,d_v)}$.*

The corollary signifies the notion of hierarchy in simulation power. Our construction of a suitable $\mathcal{U}$, as discussed after Theorem 5, ensures the existence of a computational model that can simulate any single-layer transformer attention with a given number of heads based on input $X$. We highlight that it is $X$ that dictates the width of $\mathcal{U}$, whose depth remains independent of the input. As such, the construction hinges solely on the sequence length $n$. For a sufficiently large $N$, we can inductively construct and hence prove the existence of a network, say $\mathcal{U}_{(N)}$ that can simulate arbitrary attention (or, even transformer when extended with the feed-forward component) on input having length, say $n \leq N$ – thus making it universal. The constructive proof has been provided in Appendix A.2. In the absence of a theoretical lower bound on the allowable number of heads, we can only ensure that the dependence underlying our model follows the principle of parsimony, given the natural hierarchy among simulators.

***Remark* 5.3** *(Sparsification).* Note that the $\mathrm{poly}(n)$ complexity underlying our construction of $\mathcal{U}$ stems from the definition of vanilla encoders, and does not contribute to inflation of ambient sequence length. Theorem 5 only requires $n \leq \min(d, d_v)$, which conforms to the convention in Vaswani et al. (2017). Remarkably, our approach also conforms to sparsification of pairwise token interactions, namely, methods that involve pooling to achieve appropriate compression and low-rank attentions, e.g., Linformer (Wang et al., 2020), Performer (Choromanski et al., 2021), and Sumformer (Alberti et al., 2023). This becomes crucial in mitigating the commonly encountered issue of token explosion. Linformer approximates the self-attention mechanism by a low-rank matrix —the lower rank ($k$) being prescribed based on the Johnson-Lindenstrauss Lemma —to achieve a complexity of $O(n)$. Meanwhile, Performer replaces the usual non-linearity by introducing kernels for pooling. Under Gaussian kernels, the complexity can be made as low as $O(nkd)$. Sumformers consolidate all of the above models in universally approximating sequence-to-sequence permutation-equivariant continuous functions. Our construction can be used to represent all such models as long as the underlying pooling operations are representable (see Remark 5.2).

In the purview of Lemma 3, we also extend the encoder's expressivity onto a larger class of activations. First, suppose $S$ is a sequence of length $gk$ and $\rho_g$ is a permutation, where $g, k$ are positive integers. Thus, $\rho_g(S)$ is the $g$-sorted sequence of $S$ such that $\rho_g(S)[i] \geq \rho_g(S)[i+1] \geq \cdots \geq \rho_g(S)[i+g-1]$ for all $i$ that are multiple of $g$. This permutation is often called a $\mathrm{GroupSort}$ of group size $g$. When $g = 2$, this is widely known as the $\mathrm{MaxMin}$ operation (Anil et al., 2019).

**Lemma 6.** *There exists a transformer performing* $\mathrm{MaxMin}$ *on sequence (of even length).*

*Proof.* $\mathrm{MaxMin}$ is both length-preserving and, when applied to a sequence or matrix (considered as a sequence), according to Proposition 1, yields an identical result. Algorithm 4 provides the pseudocode. Since the number of attentions does not depend on the input, it is only $O(1)$-costly. □

***Remark* 6.1** *(Approximating Lipschitz functions).* The first reason behind Lemma 6 being important is that, by representing $\mathrm{MaxMin}$, $\mathcal{U}$ can express a vector $p$-norm preserving transform, $p \geq 1$. As such, recalling that $\mathcal{U}$ also simulates affine matrix operations (multiplication), it can represent an $L$-deep feed-forward network $z^{(\ell)} := W^{(\ell)} \mathrm{MaxMin}(z^{(\ell-1)}) + b^{(\ell)}$, where $W^{(\ell)} \in \mathbb{R}^{n_\ell \times n_{\ell-1}}, b^{(\ell)} \in \mathbb{R}^{n_\ell}$, given that $||W^1||_{2,\infty} \leq 1$, $\max\{||W^{(\ell)}||_\infty\}_{\ell=2}^L \leq 1$ and $\max\{||b^{(\ell)}||_\infty\}_{\ell=1}^L \leq \infty$. In case the input vectors $z^0$ are constrained to a compact subset $Z \subseteq \mathbb{R}^{n_0}$ and $n_L = 1$, the simulated outputs are dense in $Lip_1(Z)$ (Tanielian & Biau, 2021). This presents a new proof showing that transformer encoders are universal approximators of Lipschitz and Hölder-smooth functions. Moreover, following Lemma 6, $\mathcal{U}$ exactly represents $\mathrm{ReLU}, \mathrm{LeakyReLU}$ and $\mathrm{Maxout}$ activations (Anil et al., 2019). The result extends to $\mathrm{GeLU}$-activated networks given the approximation of $\mathrm{GeLU}$ (Lipschitz-smooth with associated constant 1.0998) using $\mathrm{ReLU}$ (Feng et al. (2023), Lemma C.2).

---

**Algorithm 4:** Applying $\mathrm{MaxMin}$ sort on any sequence $S$. [see Listing 5]

1 Let $\alpha \in \{0,1\}^*$ denotes a sequence such that $\alpha[i] = \alpha[i+1] = 1$(or, 0) for $S[i] < S[i+1]$ (or, otherwise), where $i$ is even.

2 Create the attention matrix, say $\rho_2$, with diagonal blocks $\begin{pmatrix} 0 & 1 \\ 1 & 0 \end{pmatrix}$ (or, $\begin{pmatrix} 1 & 0 \\ 0 & 1 \end{pmatrix}$) depending upon $\alpha[i]$ and $\alpha[i+1]$ are both 1(or, 0) for any even $i$.

3 Return the tokens of $S$ after passing through $\rho_2$.

---

To further generalize the construction of $\mathcal{U}$, let us now work on the multi-head extension.

**Lemma 7.** *Suppose* $\circ$ *is an* $n$*-ary operation. Then, there exists a transformer* $T$ *computing* $\odot_{h=1}^H \left( \circ_{i=1}^n X_i^{(h)} \right)$ *on input* $\odot_{i=1}^n \left( \odot_{h=1}^H X_i^{(h)} \right)$, *if there is a transformer* $T^{(h)}$ *realizing the operation* $\circ$ *on input* $\odot_{i=1}^n X_i$, *where* $\odot$ *denotes concatenation.*

*Proof.* If the construction for operation $\circ$ is independent of the input, $T = T^{(h)}$ for any $h$, e.g., $\mathrm{MaxMin}$. Otherwise, we provide an explicit construction of such a transformer $T$. A transformer can implement the following operations:

- identify: A contiguous subsequence $\sigma_{i-1}\sigma_i \ldots \sigma_{i+k-1}$ from a sequence $\sigma_0 \ldots \sigma_{i-1} \ldots \sigma_{i+k-1} \ldots \sigma_{n-1}$ can be identified to produce the length-preserved sequence $0 \ldots 0\sigma_{i-1} \ldots \sigma_{i+k-1}0 \ldots 0$. The RASP code is as follows. `clip = `**`select`**`(`**`indices`**`, `**`indices`**`, `**`==`**`)`**`and`** **`select`**`(`**`indices`**`, i-1, `**`>=`**`)`**`and`** **`select`**`(`**`indices`**`, i+k-1, `**`<=`**`); `**`aggregate`**`(clip, `**`tokens`**`)`**`;`**. Line 3 (and 2 & 3) of Algorithm 2 (and 3) reminisce the property.

- shift: A cyclic permutation $\rho$ on the sequence $\sigma_0 \sigma_1 \ldots \sigma_{n-1}$ can be performed, say by an amount $t$. The RASP code is as follows `aggregate(select(indices, (indices+t)%n, ==), tokens);`. Note that applying these two operations sequentially can help shift a subsequence.

Now to construct $T$, let us first apply the identify and shift operation to permute the input sequence $\odot_{i=1}^{n} \left( \odot_{h=1}^{H} X_i^{(h)} \right)$ to $\odot_{h=1}^{H} \left( \odot_{i=1}^{n} X_i^{(h)} \right)$. The operations of $T$ would then copy the operations from $T^{(h)}$ with possible modification in `indices`, which is in $T$, is at an offset $n \times (h' - 1)$. When applied on the sequence $\odot_{h=1}^{h'-1} \left( \odot_{i=1}^{n} 0 \right) \odot_{i=1}^{n} X_i^{(h')} \odot_{h=h'+1}^{H} \left( \odot_{i=1}^{n} 0 \right)$, this would produce $\odot_{h=1}^{h'-1} \left( \odot_{i=1}^{n} 0 \right) \circ_{i=1}^{n} X_i^{(h')} \odot_{h=h'+1}^{H} \left( \odot_{i=1}^{n} 0 \right)$. Then just adding up all such $H$ sequences would produce $\odot_{h=1}^{H} \left( \circ_{i=1}^{n} X_i^{(h)} \right)$. Note that if a transformer $T^{(h)}$ requires using the FFN (e.g., the matrix multiplication), $T$ can also construct weight matrices for the FFN with possible modifications to cater only to the required portions of the produced sequence. $\square$

The purpose of this lemma is to prove that given an operation $\circ$ implementable by a transformer, another transformer can be constructed that can independently perform $\circ$, say, $H$ times, without mutual interference.

**Theorem 8.** *There exists a transformer network $\mathcal{U}$ that, on input $X$, can simulate any single-layer $H$-head transformer attention $T$ of order $(n, d, d_v)$, at its own final attention layer.*

*Proof.* Keeping congruence to the input provided to multihead attentions by Vaswani et al. (Subsection 3.2.2), we assume that the characterizing matrices have been stacked one after another, i.e., $\left( \odot_{h=1}^{H} X^{(h)} \right) \odot \left( \odot_{h=1}^{H} A^{(h)} \right) \odot \left( \odot_{h=1}^{H} V^{(h)} \right)$, where, $X^{(h)} \in \mathbb{R}^{n \times d}, A^{(h)} \in \mathbb{R}^{d \times d}$ and $V^{(h)} \in \mathbb{R}^{d \times d_v}$ is the input to network $\mathcal{U}$. Thus, the construction of $\mathcal{U}$ follows from Lemma 7 and Lemma 2-4. $\square$

Evidently, the result also extends to the entire transformer.

**Corollary 8.1.** *There exists a transformer network $\mathcal{U}$ that, on input $X$, can simulate any single-layer $H$-head transformer $T$ of order $(n, d, d_v, d_1, d_2)$, at its own final attention layer.*

***Remark* 8.1.** With the simulation of multi-head transformers, an architecture can be realized through explicit construction for the problems which are known to be learnable using single-layer multi-head transformers, e.g., $k$-PARITY. Note that in terms of RASP, a residual connection is only an elementwise sum. Accordingly, for the task of recognizing the language PARITY, the two-layer $\mathrm{softmax}$ encoder architecture proposed by Chiang & Cholak can be realized using average hard attention by employing two serially connected $\mathcal{U}$ networks.

# 6 CONCLUSION

We present for the first time an exact, data-agnostic construction of a universal simulator that replicates the behavior of single-layer transformer encoders, including multi-head attention and non-linear feed-forward components. Central to our construction is the implementation of key linear algebraic operations and a wide-range of activation functions, all within the constant-depth constraint of transformer architecture. Our results demonstrate that while such structure precludes simulation of arbitrary encoder configurations, a hierarchical construction exists wherein simulators of higher-order subsume lower-order models. Crucially, extending this to multi-head attention as in Theorem 8 ensures the existence of a universal simulator $\mathcal{U}$. As an obvious extension of this work and backed by the Turing completeness of transformers, one may investigate the construction of an analogous mechanism involving an encoder-decoder-based model to simulate an arbitrary transformer. By constructing average-hard attention-based models that exactly replicate $\mathrm{softmax}$-activated attention, we show that algorithmic approximations of problems previously believed to be learnable only through training, such as $\mathrm{Match}_2$ and $k$-PARITY. This rigorously shifts the boundary between empirical approximation and formal simulation in attention-based models. The development of RASP compilers such as Tracr (Lindner et al., 2023) and ALTA (Shaw et al., 2025) presents a promising avenue for obtaining realized weights corresponding to the proposed network $\mathcal{U}$. However, challenges towards a complete implementation of the RASP framework still exist in either of these compilers. Along such a line, learning the algorithmically developed transformers and $\mathcal{U}$ via existing optimization-based methods and coming up with a convergence criterion may be considered as future work.

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

## A APPENDIX

### A.1 RASP CODES FOR IMPLEMENTING THE NETWORK $\mathcal{U}$

This section will provide the deferred proof and RASP codes referenced in Section 4.

*Proof of Proposition 1.* We assume that performing basic arithmetic operations is supported while doing this conversion. We would apply induction on $n$ to prove the proposition. Avoiding the trivial case when $n = 1$, let us consider a two-dimensional array $A$ having $m$ rows and $m$ columns. Note that any element $A[i_0, i_1]; i_l \in \{0, 1, \ldots, m_l - 1\}, l \in \{0, 1\}$ can be accessed from the one-dimensional array $A'$ of size $m_0 m_1$ using the elementary index calculation $m_0 i_0 + i_1$. We assume that there exists a one-dimensional representation $A'$ for the $n$-dimensional array $A$. To construct an one-dimensional representation $A''$ for an $n + 1$-dimensional array $A$, let $A'_j$ denotes the one-dimensional representation of the $n$-dimensional array $A[j], j \in \{0, 1, \ldots, m_0 - 1\}$. Thus, concatenation of all such $A'_j$, say $A''$ is the one-dimensional representation of $A$ such that the element $A[i_0, i_1, \ldots, i_n]$ can be accessed in $A''$ using index $\prod_{l=1}^{n} m_l i_0 + k$, where $k$ is the index of the element in $A'_{i_0}$. $\square$

```
1 def Transpose_r(){
2   r, c = 3, length/3;
3   reflectedIndices = (indices%r)*c + ((indices-indices%r))/r;
4   reflect = select(indices, reflectedIndices, ==);
5   return aggregate(reflect, tokens_int);
6 }
```

Listing 1: Transposing a matrix of order 3 implementing Algorithm 1. Note that the transpose operation is a length-preserving operation.

```
1 def softmaxrect_r(){
2   r, c = 3, length/3;
3   exp = (2.73^tokens_float);
4   sel1, sel2, sel3 =(select(indices, c*0+c, <) and select(c*0+c,
       indices, >)), (select(indices, c*0+c, >=) and select(indices, c*1+c,
        <) and select(c*1+c, indices, >) and select(c*0+c, indices, <=)), (
       select(indices, c*1+c, >=) and select(indices, c*2+c, <) and select(
       c*2+c, indices, >) and select(c*1+c, indices, <=));
5   denom1, denom2, denom3 = c*aggregate(sel1, exp), c*aggregate(sel2,
       exp), c*aggregate(sel3, exp);
6   denom = (denom1+denom2+denom3);
7   return exp/denom;
8 }
```

Listing 2: Applying $\mathrm{softmax}$ (a length-preserving operation) on matrix $A$ of order 3 implementing Algorithm 2.

```
1 def Matmul_3dot4(){
2   k = length/(3+4);

3   one_a, one_b, two_a, two_b, three_a, three_b, four_b = indices%k, (
       indices%k)*4+3*k, (indices%k)+1*k, (indices%k)*4+3*k+1, (indices%k)
       +2*k, (indices%k)*4+3*k+2, (indices%k)*4+3*k+3;

4   one_sa, one_sb, two_sa, two_sb, three_sa, three_sb, four_sb = select(
       indices, one_a, ==), select(indices, one_b, ==), select(indices,
       two_a, ==), select(indices, two_b, ==), select(indices, three_a, ==)
       , select(indices, three_b, ==), select(indices, four_b, ==);

5   oneone_ab, onetwo_ab, onethree_ab, onefour_ab, twoone_ab, twotwo_ab,
       twothree_ab, twofour_ab, threeone_ab, threetwo_ab, threethree_ab,
       threefour_ab = aggregate(one_sa, tokens_int)*aggregate(one_sb,
       tokens_int), aggregate(one_sa, tokens_int)*aggregate(two_sb,
```

```
      tokens_int), aggregate(one_sa, tokens_int)*aggregate(three_sb,
      tokens_int), aggregate(one_sa, tokens_int)*aggregate(four_sb,
      tokens_int), aggregate(two_sa, tokens_int)*aggregate(one_sb,
      tokens_int), aggregate(two_sa, tokens_int)*aggregate(two_sb,
      tokens_int), aggregate(two_sa, tokens_int)*aggregate(three_sb,
      tokens_int), aggregate(two_sa, tokens_int)*aggregate(four_sb,
      tokens_int), aggregate(three_sa, tokens_int)*aggregate(one_sb,
      tokens_int), aggregate(three_sa, tokens_int)*aggregate(two_sb,
      tokens_int), aggregate(three_sa, tokens_int)*aggregate(three_sb,
      tokens_int), aggregate(three_sa, tokens_int)*aggregate(four_sb,
      tokens_int);

6   sel_one, sel_two, sel_three, sel_four, sel_five, sel_six, sel_seven,
      sel_eight, sel_nine, sel_ten, sel_eleven, sel_twelve = select(
      indices, k, <) and select(0, indices, ==), select(indices, k, <) and
       select(1, indices, ==), select(indices, k, <) and select(2, indices
      , ==), select(indices, k, <) and select(3, indices, ==), select(
      indices, k, <) and select(4, indices, ==), select(indices, k, <) and
       select(5, indices, ==), select(indices, k, <) and select(6, indices
      , ==), select(indices, k, <) and select(7, indices, ==), select(
      indices, k, <) and select(8, indices, ==), select(indices, k, <) and
       select(9, indices, ==), select(indices, k, <) and select(10,
      indices, ==), select(indices, k, <) and select(11, indices, ==);

7   matmul = k*(aggregate(sel_one, oneone_ab)+aggregate(sel_two,
      onetwo_ab)+aggregate(sel_three, onethree_ab)+aggregate(sel_four,
      onefour_ab)+aggregate(sel_five, twoone_ab)+aggregate(sel_six,
      twotwo_ab)+aggregate(sel_seven, twothree_ab)+aggregate(sel_eight,
      twofour_ab)+aggregate(sel_nine, threeone_ab)+aggregate(sel_ten,
      threetwo_ab)+aggregate(sel_eleven, threethree_ab)+aggregate(
      sel_twelve, threefour_ab));
8     return matmul;
9 }
```

Listing 3: Multiplying two matrices of shape $3 \times \cdot$ and $\cdot \times 4$ implementing Algorithm 3.

```
1 def ReLU(){
2   return (0 if tokens<0 else tokens);
3 }
```

Listing 4: Implementation of ReLU.

```
1 def MaxMinSort(){
2   MaxSel = select(indices, indices, ==);
3   MinSel = select(indices, indices+1, ==) and select(1, indices%2+1, ==
      );
4   MaxminusMin = aggregate(MaxSel, tokens) - aggregate(MinSel, tokens);
5   reqFlip = 1 if MaxminusMin<0 else 0;
6   reqFlip = reqFlip + aggregate(select(indices+1, indices, ==), reqFlip
      );
7   revby2 = aggregate(select(0, indices%2, ==), 1) + aggregate(select(1,
       indices%2, ==), -1);
8   flip = select(indices, indices+revby2, ==);
9   sorted = reqFlip*aggregate(flip, tokens) + (1-reqFlip)*aggregate(
      select(indices, indices, ==), tokens);
10  return sorted;
11 }
```

Listing 5: Implementation of MaxMin sort realizing Algorithm 4.

**Complexity of individual operations.** To illustrate the complexities in accordance with the discussion in Section 5, we present the following analysis on the RASP codes. Listing 1 generates an attention matrix that maps the indices to their transposed position and then passes the tokens to

get permuted accordingly. This requires a single non-trivial attention layer and a preceding layer to compute n and reflectedIndices. The reflectedIndices is in fact the permutation $\rho$ as defined in line 2 of Algorithm 1. For implementing the operation softmax in Listing 2, the principal attention layer has a width of 3, computing the row sums. Similarly, the function matrix multiplication in Listing 3 requires two non-trivial layers and an opening layer for the calculation of several index manipulations. Line 3 is performing necessary index calculations for implementing lines 2, 3 of the respective algorithm. The second attention layer corresponds to lines 4-5 and thus has a width of seven, while the third layer, corresponding to lines 6-7, has width twelve. For easy understanding, we have presented the keyword **tokens** in the proofs; however, following the RASP semantics, we have used **tokens_float** (or **tokens_int**) while dealing with numerals. A standard construction of UTM stores the transitions of an input TM using some delimiter (mostly a predefined number of 0s). One may get intimidated to apply the same to delimit the rows of a matrix when presented as a sequence using Proposition 1. Though that would help to count the rows and thus columns, the architecture of transformers inhibits us from directly looping on the rows or columns to bypass the explicit construction of the **select-aggregate** pairs (e.g., the three selectors sel1, sel2, and sel3 in Listing 2).

## A.2   PROOF OF COROLLARY 5.2

We prove this by construction. Let $T$ be a single-layer transformer with attention matrix $A \in \mathbb{R}^{e \times e}$, value matrix $V \in \mathbb{R}^{e \times e_v}$, and input $X \in \mathbb{R}^{m \times e}$. To simulate $T$ using $\mathcal{U}_{(n,d,d_v)}$, we proceed as follows:

*Step 1* (Input Embedding). Pad the input $X$ to $\widetilde{X} \in \mathbb{R}^{n \times d}$ via zero-padding and block-diagonal extension:

$$\widetilde{X} = \begin{bmatrix} X & \mathbf{0}_{m \times (d-e)} \\ \mathbf{0}_{(n-m) \times e} & \mathbf{0}_{(n-m) \times (d-e)} \end{bmatrix},$$

where $\mathbf{0}_{p \times q}$ denotes a zero matrix of size $p \times q$.

*Step 2* (Attention/Value Matrix Embedding). Similarly, embed $A$ and $V$ into higher-dimensional spaces:

$$\widetilde{A} = \begin{bmatrix} A & \mathbf{0} \\ \mathbf{0} & \mathbf{I}_{d-e} \end{bmatrix}, \quad \widetilde{V} = \begin{bmatrix} V & \mathbf{0} \\ \mathbf{0} & \mathbf{0} \end{bmatrix},$$

where $\mathbf{I}_k$ is the $k \times k$ identity matrix. The identity block ensures that padded dimensions do not interfere with the computation.

*Step 3* (Simulation). By construction, $\mathcal{U}_{(n,d,d_v)}(\langle \widetilde{A}, \widetilde{X} \rangle)$ computes:

$$\text{softmax}\left( \frac{\widetilde{X}\widetilde{A}\widetilde{X}^{\top}}{\sqrt{d}} \right) \widetilde{X}\widetilde{V},$$

which reduces to the original computation $T(X)$ in the upper-left $m \times e_v$ block. The padded dimensions contribute only trivial linear transformations (due to $\mathbf{0}$ and $\mathbf{I}$ blocks), leaving the simulation exact.

## A.3   ALTERNATIVE CONSTRUCTION FOR $\mathcal{U}$

Before the discussion for constructing $\mathcal{U}$, let us implement another elementary matrix operation – inversion.

**Lemma 9.** *There exists a transformer that can invert a non-singular matrix A of rank 3.*

*Proof.* Given a matrix $A$ and its mapped sequence from Proposition 1, the inversion operation is also length-preserving. We shall adopt an analytical framework for inverse computation, utilizing the fundamental operations of matrix cofactor, determinant, and adjugate. The final transposition step can be derived through the application of Lemma 2. The RASP pseudocode is presented in Algorithm 5.

The RASP code for finding cofactor and determinant has been provided in Listing 6, 7.

---

**Algorithm 5:** Finding Cofactor and Determinant of square matrix $A$ of rank 3.

1  $r \leftarrow$ rank of $A$.
2  Identify the four sequences of indices of length $r^2$, say, $\alpha, \beta, \gamma$ and $\delta$ such that the indices $\alpha[i], \beta[i], \gamma[i]$ and $\delta[i]$ contain the tokens responsible for computing the cofactor corresponding to the token at index $i$ of matrix $A$.
3  Attend to the tokens $P, Q, R$ and $S$ at the sequences of indices $\alpha, \beta, \gamma$ and $\delta$, respectively.
4  Then the cofactor of matrix $A$, $M_A$ is element-wise operation $P \times Q - R \times S$.
5  Multiply the tokens of $A$ and that of $M_A$, element-wise.
6  Create a mask that can attend to only the first $r$ position.
7  And apply the mask to the multiplied result from line 5.
8  Add all the elements in the resultant sequence from line 7 and obtain the determinant of $A$, a sequence of length $r^2$ filled with $|A|$.

---

The expression $M_A/|A|$ (or, `Cofactor()/Det(Cofactor())(A)`) now yields the transpose of the adjugate $\mathrm{Adj}(A)$. Subsequently, applying the transposition operation to the adjugate results in the desired matrix inverse $A^{-1}$. □

```
1  def Cofactor(){
2    n = length^0.5;
3    i,j = (indices-indices%n)/n, indices%n;

4    idx1, idx2, idx3, idx4 = (i+1)%n, (j+1)%n, (i+2)%n, (j+2)%n;
5    one, two, three, four = idx3*n+idx4, idx1*n+idx2, idx3*n+idx2, idx1*n
       +idx4;

6    sel_one, sel_two, sel_three, sel_four = select(indices, one, ==),
       select(indices, two, ==),select(indices, three, ==), select(indices
       , four, ==);
7    P, Q, R, S = aggregate(sel_one, tokens), aggregate(sel_two, tokens),
       aggregate(sel_three, tokens), aggregate(sel_four, tokens);
8    cofactor = P*Q-R*S;
9    return cofactor;
10 }
```

Listing 6: Finding Cofactor of a Matrix as a part of implementing Algorithm 5.

```
1    def Det(Cofactor){
2      n = length^0.5;
3      mask = select(indices, n, <) and select(indices, indices, ==);
4      det = length*aggregate(full_s, aggregate(mask, (tokens*Cofactor))
         );
5      return det;
6    }
```

Listing 7: Finding Determinant of a Matrix as a part of implementing Algorithm 5.

Consider a transformer network $\mathcal{U}$ that adapts its parameters in the final attention layer depending on the input parameters $A, V$, and $X$, where $A$ and $V$ are the non-singular characterizing matrices of an attention $T$. Although such a construction does not solely satisfy the motivation as depicted in Figure 1, it may be worth an attempt to explore the expressive power of a transformer implemented using RASP.

From the fact that $\det(M_1 M_2) = \det(M_1)\det(M_2) \neq 0$ when matrices $M_1$ and $M_2$ are both non-singular, it is straightforward to see that matrix $M_1 M_2$ is also non-singular. Now, we will construct $\mathcal{U}$, which will take input a sequence $X, A$ and $V$ such that the final attention layer, say $L$ receives input $XAV$. Thus, to simulate $T$ on input $X$, the attention and value matrices at layer $L$ of $\mathcal{U}$ must be $\left((AV)^\top V\right)^{-1}$ and $(AV)^{-1} V$ respectively, so that it produces:

$$\sigma\left((XAV)\left((AV)^\top V\right)^{-1}(XAV)^\top\right)(XAV)(AV)^{-1}V \qquad (4)$$

As long as the characterizing matrices of $T$ are non-singular and of rank 3, the attention and value matrices of $\mathcal{U}$ can be realized through a series of sequential operations implementable using the previous lemmas.

The function `Cofactor()`, a non-trivial attention layer with four heads representing the agents to pick the four sequences of elements involved to calculate the respective cofactor, and a preceding layer responsible for index calculation. The other one `Det()` calculating determinant takes the function `Cofactor()` as an argument, thus giving rise to two additional attention layers where the first one attends to the first three indices and the following layer is a trivial one doing the multiplication with **length**. However, while inverting a matrix, we may combine some arithmetic operations, mostly taking place in the first layer of attention of the aforementioned functions, which will help us to get a five-layer transformer having a width of four (see Figure 2). The existing implementations of calculating inverse (e.g., Giannou et al. (2023)) involve Newton's iterative formula. The constant-depth (13-deep, 1-wide) transformers only approximate the solution (up to $T$ steps), and fundamentally, rely on a computational framework that is neither entirely transformer-based (as they use `for`) nor the classical computational paradigm (as they use transformers). On the other hand, it is important to discuss the constructional challenges with matrix inversion in RASP. Despite producing an exact solution, computing the cofactor of a rank-$(k+1)$ non-singular matrix depends on that of precisely $(k+1)^2$ rank-$k$ matrices. Achieving this inherent *recursive* computation for arbitrary $k$ using a constant-depth architecture such as a transformer is not amenable.

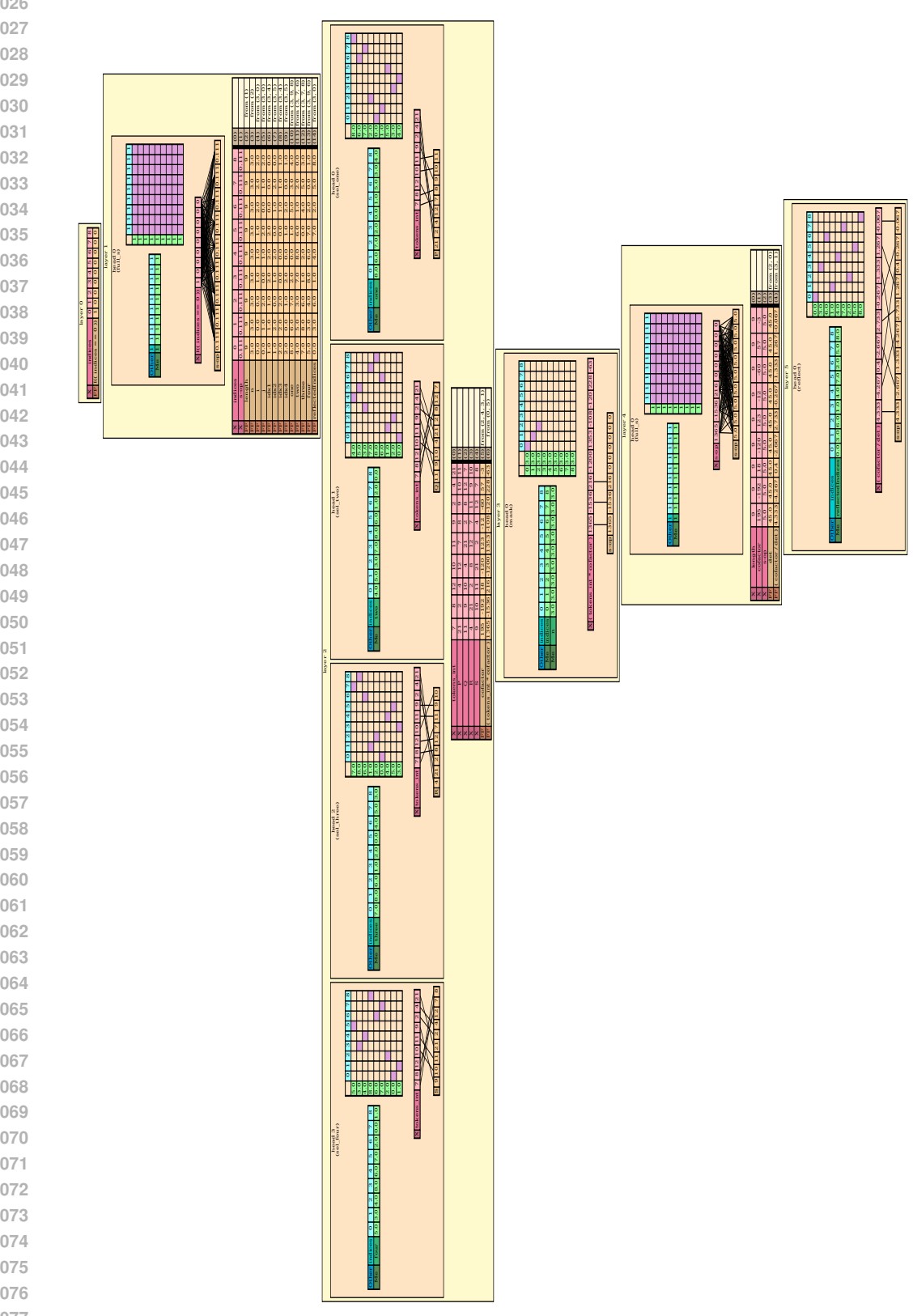

Figure 2: Constructed transformer inverting a non-singular matrix $A = \begin{pmatrix} 7 & 8 & 12 \\ 10 & 11 & 9 \\ 2 & 4 & 21 \end{pmatrix}$. For a clear view, see https://anonymous.4open.science/r/TMA/Inverse.pdf.

