# OpenReview forum: "On the Existence of Universal Simulators of Attention"
_ICLR.cc/2026/Conference — ICLR 2026 Conference Withdrawn Submission_

### Official Review · Reviewer_FRte · 2025-10-16

**Soundness:** 2
**Presentation:** 1
**Contribution:** 2
**Rating:** 2
**Confidence:** 4

**Summary:**

The paper under review attempts to show that there exists a ``universal transformer network’’ for transformer encoders. What the result seems to say is that there exists a transformer that, given a sequence of vectors x_1, …, x_n \in mathbb{R}^d, and matrices K, Q, V, performs the attention layer on input (x_1, .., x_n) with K, Q, V as key, query, and value matrices.

The construction uses a programming language RASP that is known (?) to be simulatable by an average hard-attention transformer. As a consequence the paper claims to solve an open problem by showing that softmax transformers can be simulated by average-hard transformers.

**Strengths:**

The paper attempts to resolve an open question of whether softmax transformers can be simulated by AHAT.

**Weaknesses:**

I believe that the main results are neither stated nor proved with necessary rigor. In Proposition 1, what does it mean ``represented’’? In Lemmas, 2,3,4 how the input matrices are given to the transformer and how they are split into tokens? What is the input length? What are quantifiers over r, k? In Theorem 5, what does it formally mean to simulate? Are Key, Query and Value matrices of T are parts of the input to U?

The proofs of Lemmas 2,3,4 are given in the form of a RASP code without any comments. A pseudo code can be used to illustrate the proof, not to replace it.

RASP language has not been defined in the paper. It makes it inconvenient to verify the proofs. Is there actually any result in the literature that RASP can be simulated by AHAT? It is introduced in Weiss, Goldberg and Yahav but I have not found any formal results. As I understand, Yang and Chiang (Counting Like Transformers: Compiling Temporal Counting Logic Into Softmax Transformers) show this result for a modification of RASP called C-RASP, and for softmax transformers.

Finally, the paper is written in a confusing language. Some examples

``By dimension of a multi-dimensional array M, we signify the number of axes referred in M’’ signify <- mean? By ``the dimension’’? I guess, if you are trying to define the number of dimensions  of an array, you cannot use the word ``axes’’ which is essentially the same thing.


``the induced (n−1)-dimensional array hosted from the index of the introductory axis in M’’ - I don’t understand this phrase

Line 173 ``where |*| implies’’ denotes?

Line 181 ``formalizes the same’’ – ``formalizes the above’’
 etc.

**Questions:**

no questions

---

### Official Review · Reviewer_h1Gc · 2025-10-30

**Soundness:** 2
**Presentation:** 2
**Contribution:** 3
**Rating:** 2
**Confidence:** 5

**Summary:**

The paper reports a RASP construction for simulating one attention layer on in input string. The main result is that we can construct a RASP program U that, on input T, X, simulates T(X). Here, T is a single multi-head attention layer. Since RASP is a model of transformer computation, this result serves as a kind of "universal simulator result": a transformer U that can simulate other transformers, similar in spirit to a universal Turing machine. However, the theorem statements and proofs of this claim are not entirely rigorous.

**Strengths:**

1. The goal of the paper (constructing a transformer that simulates transformers) is quite cool and would make an interesting addition to the analysis of transformer expressivity.
2. The high-level argument looks reasonable, though the detailed formal analysis should be made more rigorous.

**Weaknesses:**

### Rigor Issues with Theoretical Results

My major concern with this paper is that there are many rigor issues in the theorem statements and proofs.

As a first minor point, Lemma 2 onwards make a statement about a "transformer", but really they are about the existence of RASP programs, which correspond to a specific transformer model.

More importantly, in Proposition 1 and Lemma 2 onwards, it's unclear how the input and output are represented in these constructions, as transformer inputs, outputs, and representations are sequences. Clarifying the details of the representations is important since you will be composing these operations in Theorem 5 to simulate U, and it's important that they fit together properly. The paper should have more rigorous lemma statements that clarify how all these operations can be appropriately composed within a transformer.

Algorithms 1-3 are pseudocode, which is very far from both transformers and RASP. E.g., in algorithm 1, line 3, it doesn't say how you construct an attention head that maps from [i, j] to [j, i]. Clarifying the representational details mentioned above and adjusting the algorithms to refer to that would make them more rigorous. Even better, you should actually write algorithms 1-3 in RASP so that it's clear what version of RASP (i.e., what primitives) are required.

Theorem 5 and 8 are confusingly worded: there exists a transformer U, that, on input X, simulates any one-layer transformer T. This is vacuously true: just set T = U. You actually mean U takes input <T, X> and simulates T(X). But, related to the issues above, it's not made clear how T should be encoded in the input to U.

What does it mean to implement the operation softmax in Lemma 3? The output of softmax is irrational, so you are approximating it. But it's not clear in what sense the approximation holds.

The results about MaxMin and Lipschitz continuous functions are hard to verify. The authors should clarify how the input and output are represented (if they are real numbers?), make rigorous the notion of approximation that can be achieved, and analyze the size of the transformer that is needed (e.g., as a function of the Lipschitz constant). In addition to clarifying these details, it would helpful to discuss how universal approximation result sits with the known result that any poly-size transformer encoder with poly-precision can only compute functions (indicator functions for formal languages) in TC0.
### Clarify the Version of RASP Used

The original RASP is not a well-defined computational model. It's important that RASP constructions don't hide arbitrary computation in their elementwise operations, which is why later works have taken pains to define different fully defined RASP variants like B-RASP, C-RASP, etc. It's not very clear what model of RASP is assumed for the results in this paper, but clarifying that it's some simple flavor of RASP would strengthen the appeal of the results.
### Claim of Match2 Novelty is Overstated

> As a consequence, we can construct an AHAT for problems such as Match2, known until now to be only learnable using single-layer single-head SMATs.

Without going through your construction, it's fairly straightforward to implement match2 (e.g., mod 10) with AHAT: each x_i has attends to its left to see if it can find x_j = k, where k is the unique value -x_i mod 10. This is a direct one-layer AHAT construction that doesn't require going through your simulation. More generally, AHAT constructions are generally easier than SMAT constructions, so I'm not sure if there are clear examples where simulating SMAT with AHAT will let us solve something we couldn't otherwise solve.

**Questions:**

> has largely been data-driven, offering only probabilistic rather than deterministic guarantees.

Data-driven suggests no guarantees at all?

> These models have demonstrated the ability to learn from tasks and function as simulators of a broad range of computational architectures.

What exactly do you mean by simulators here?

Unclear what value the discussion of parity in the introduction provides. First, you should point out that the different results about parity are due to different models of transformers (hard vs. soft attention). Second, you should clarify why you're bringing up parity, match2, etc., and how it relates to your research question and results.

Split discussion of RASP in Section 3 into its own subsection, since this is a core part of the paper

---

### Official Review · Reviewer_s3rd · 2025-11-01

**Soundness:** 2
**Presentation:** 2
**Contribution:** 2
**Rating:** 4
**Confidence:** 3

**Summary:**

This paper presents data-agnostic construction of a universal simulator that replicates the behavior of single-layer transformer encoders. The authors provide mathematical analysis on such claim.

**Strengths:**

This paper focuses on an interesting direction of understanding the learnability and expressivity of transformers.

**Weaknesses:**

* The theoretical results are hard to parse. It would be better if the authors provide graphical illustration on what they proved and why it makes sense.

* No empirical results supporting the theory.

**Questions:**

None

---

### Official Review · Reviewer_fgAY · 2025-11-04

**Soundness:** 2
**Presentation:** 2
**Contribution:** 2
**Rating:** 2
**Confidence:** 4

**Summary:**

This paper’s writing requires significant improvement. I found it difficult to even identify the stated contributions.
 To my understanding, the paper demonstrates that, given access to RASP commands, specifically any point-wise function, aggregation, and selection, one can reconstruct any attention layer, provided the model’s input and the weight matrices of the targeted layer are given in-context. The authors further claim that since each of these RASP commands can be translated into transformer layers, a transformer can therefore emulate any single attention layer.

**Strengths:**

I think the scope of this work is meaningful and interesting;  finding which is the minimal set of operations needed to reconstruct attention. This work takes a step towards this direction by showing that RASP, a programming language designed such that each of its operations can be translated to transformer layers, is actually sufficient to reconstruct one layer of encoder based attention. Furthermore, the authors show how RASP can also reconstruct other type of attention mechanisms like Linformer and Linear Attention.

**Weaknesses:**

1. The main contribution of the paper is not clear. According to the statements made in the paper, it seems that the main contribution is the construction of the universal simulator $U$, composed of transformer encoder layers, that is able to simulate any attention layer in-context. However, this is straightforward using the results of [1], which shows any matrix multiplication, transposition and of course softmax is feasible with encoder based architectures. While the authors claim that they improve upon [1], it is unclear which is this improvement. Note that in line 065-066  [1] requires dxd^2 input, which I believe is consistent with the requirements in this work). I suggest to the authors to make a table in which clearly state the number of layers, width etc their final model (or each individual model) has and how this improves upon [1].

     Thus, if the authors consider the main contribution of the paper the in-context simulation of attention with an encoder-based model, I think the novelty here is limited. As mentioned before, I find more interesting the emulation of attention using RASP only. However, I think the paper would be improved by adding an explicit justification on the choice of RASP and **how many** layers and width the construction requires, meaning how many commands the RASP program has when the input matrix is $d\times n$. The reported Table 1 does not clearly state what cost is.

2. The flow of the paper should be significantly improved. I would suggest the authors to clearly state which are the contributions of the paper and the main points they want to make. At its current form this paper is confusing. For example, even in the abstract the sentence: "can transformer architectures exactly simulate an arbitrary attention mechanism, or in particular, the underlying operations?" reads as trivial. A transformer can simulate attention, since it contains attention module within its architecture. One way to improve the phrasing would be that can it simulate in-context.

   The paper makes the claim of exact construction throughout . I think this claim is incorrect. As an example, notice that in Listing 2 the operation 2.73^tokens_float is used. Even tough, RASP states any point-wise operation - what is actually meant is any operation that the non-linear layers can **approximate**. Notice that  this means that exponentials would introduce some error when translated to transformer layers. This has also been analyzed in [1].

3. Related to the previous point, I think the authors should actually construct using RASP the transformer layers to show that the simulation is exact, in case this is possible. Then show that the output of the attention layer is indeed identical to the constructed model.

[1]: Giannou, A., Rajput, S., Sohn, J., Lee, K., Lee, J.D. &amp; Papailiopoulos, D.. (2023). Looped Transformers as Programmable Computers. <i>Proceedings of the 40th International Conference on Machine Learning</i>, in <i>Proceedings of Machine Learning Research</i> 202:11398-11442 Available from https://proceedings.mlr.press/v202/giannou23a.html.

**Questions:**

1. Could the authors clearly state the contributions of the paper compared to previous work?

2. Could the authors comment on the exact construction claims made throughout the paper? Not only the example mentioned above.

3. I would suggest also improving the flow of the text and creating a small experiment (as described above) to support their claims.

---

### Note · Authors · 2025-11-21

I have read and agree with the venue's withdrawal policy on behalf of myself and my co-authors.